# New Transcriptomic Biomarkers of 5-Fluorouracil Resistance

**DOI:** 10.3390/ijms24021508

**Published:** 2023-01-12

**Authors:** János Tibor Fekete, Balázs Győrffy

**Affiliations:** 1Research Center for Natural Sciences, Cancer Biomarker Research Group, Institute of Enzymology, Magyar Tudósok Krt. 2, H-1117 Budapest, Hungary; 2National Laboratory for Drug Research and Development, Magyar Tudósok Krt. 2, H-1117 Budapest, Hungary; 3Department of Pediatrics, Semmelweis University, H-1094 Budapest, Hungary; 4Department of Bioinformatics, Semmelweis University, H-1094 Budapest, Hungary

**Keywords:** pharmacology, proliferation, receiver operator characteristics, RNAseq, gene expression

## Abstract

The overall response rate to fluoropyrimidine monotherapy in colorectal cancer (CRC) is limited. Transcriptomic datasets of CRC patients treated with 5-fluorouracil (5FU) could assist in the identification of clinically useful biomarkers. In this research, we aimed to analyze transcriptomic cohorts of 5FU-treated cell lines to uncover new predictive biomarker candidates and to validate the strongest hits in 5FU-treated human colorectal cancer samples with available clinical response data. We utilized an in vitro dataset of cancer cell lines treated with 5FU and used the reported area under the dose–response curve values to determine the therapeutic response to 5FU treatment. Mann–Whitney and ROC analyses were performed to identify significant genes. The strongest genes were combined into a single signature using a random forest classifier. The compound 5-fluorouracil was tested in 592 cell lines (294 nonresponders and 298 responders). The validation cohort consisted of 157 patient samples with 5FU monotherapy from three datasets. The three strongest associations with treatment outcome were observed in *SHISA4* (AUC = 0.745, *p*-value = 5.5 × 10^−25^), *SLC38A6* (AUC = 0.725, *p*-value = 3.1 × 10^−21^), and *LAPTM4A* (AUC = 0.723, *p*-value = 6.4 × 10^−21^). A random forest model utilizing the top genes reached an AUC value of 0.74 for predicting therapeutic sensitivity. The model correctly identified 83% of the nonresponder and 73% of the responder patients. The cell line cohort is available and the entire human colorectal cohort have been added to the ROCPlot analysis platform. Here, by using in vitro and in vivo data, we present a framework enabling the ranking of future biomarker candidates of 5FU resistance. A future option is to conduct an independent validation of the established predictors of resistance.

## 1. Introduction

Globally, colorectal cancer (CRC) accounts for 10% (1.9 million) of the overall cancer incidence and 9.4 % (935,000) of deaths caused by cancer. In 2020, it placed third in incidence and second in mortality among all cancer types [1]. In terms of pathogenesis, approximately 70% of the diagnosed cases are sporadic, 25% are familial, and the remaining 5% are inherited [2]. The primary treatment option is surgery, but in advanced stages, systemic chemotherapy is the standard treatment. The adjuvant therapy for resected CRC patients without metastases includes capecitabine, oxaliplatin, 5-fluorouracil (5FU), and leucovorin [3]. Treatment for metastatic patients is based on 5FU, oxaliplatin, and irinotecan, and, depending on the genetic profile, may also involve biological therapies such as those targeting the VEGF or EGFR pathways [4].

Predictive biomarkers can be used to predict the outcome of a therapeutic intervention and can provide information on the expected success rate of a particular therapy, the existence of therapeutic resistance, or the development of a serious adverse reaction. Although biomarkers can contribute to personalized treatment with improved outcomes, the number of predictive biomarkers in colorectal cancer is still limited. One of the few biomarkers that influence therapeutic decision-making is microsatellite instability (MSI). The 5FU monotherapy treatment is not effective in patients with high MSI, but oxaliplatin-based treatment can offer a therapeutic benefit. Another biomarker affecting therapeutic decisions is the presence of KRAS/NRAS mutations, which decrease the efficacy of EGFR therapies [5]. An important predictive biomarker for avoiding serious adverse reactions to 5FU-based treatment is the test for dihydropyrimidine dehydrogenase (DPD) deficiency in patients [6]. 

Both 5FU and capecitabine are fluoropyrimidines, which are members of the class of agents known as antimetabolites. As their chemical structure shares a number of common traits with the substrate of enzymes essential for DNA synthesis, antimetabolites can disrupt the DNA structure and ultimately lead to tumor cell death. Besides colorectal cancer, fluoropyrimidines are also used to treat breast, head and neck, ovarian, and gastrointestinal tumors, as well as basal cell carcinomas. The anti-tumor effects of fluoropyrimidine analogs are complex and involve three different mechanisms of action: (a) the 5FU metabolite fluorodeoxyuridine monophosphate (FdUMP) inhibits thymidylate synthase, which is essential for thymidine synthesis, (b) fluorouridine triphosphate (FUTP) is incorporated into RNA, and (c) fluorodeoxyuridine triphosphate (FdUTP) is incorporated into DNA resulting in DNA strand breaks [7]. 

The overall response rate to fluoropyrimidine monotherapy in colorectal cancer is limited and is in the range of 10% to 15% [8]. A higher response rate of up to 45–50% can be achieved with additional oxaliplatin or irinotecan treatment; however, in these cases, toxicity will also increase [9]. The incidence of CRC is rising and 5FU is considered a key drug; therefore, there is an urgent need to be able to identify patients who may benefit from 5FU-based therapies. Transcriptomic datasets profiling tumors of CRC patients treated with 5FU could provide help with making a significant advancement in this area.

In this study, we aimed to analyze transcriptomic cohorts of 5FU-treated cell lines to uncover new predictive biomarker candidates and validate the strongest hits in 5FU-treated colorectal human samples with clinical information and response data. Our overall goal was not only to identify single genes that could serve as biomarkers in patients treated with fluoropyrimidine but also to combine the top candidates into a single predictive tool by employing machine learning. 

## 2. Results

### 2.1. Features Significant in the In Vitro Dataset

The compound 5-fluorouracil was tested in 907 cell lines, with a minimum screening concentration of 0.125 μM and a maximum concentration of 32 μM. Based on the reported AUDRC values, 294 were categorized as nonresponders (AUDRC range: 0.931–0.991), 298 as responders (AUDRC range: 0.099–0.801), and 315 (AUDRC between 0.931 and 0.801) were excluded from the analysis. The most sensitive solid tumor cell lines were found to be PSN1 (pancreas cancer), CAL148 (breast cancer), and JHU011 (head and neck cancer), whereas the most resistant cell lines were LN18 (glioblastoma), HEC1 (uterus cancer), and ASH3 (cervix cancer) (Table 1).

All available genes (n = 15,791) were tested by the Mann–Whitney U test and ROC analysis, and we found statistically significant differences between nonresponders and responders in 2484 genes. The strongest associations with treatment outcome were observed in *SHISA4* (shisa family member 4 gene, n = 592, ROC AUC: 0.745, Mann–Whitney U test *p*-value: 5.5 × 10^−25^), *SLC38A6* (solute carrier family 38 member 6 gene, ROC AUC = 0.725, *p*-value = 3.1 × 10^−21^), and *LAPTM4A* (lysosomal protein transmembrane 4 alpha gene, ROC AUC = 0.721, *p*-value = 1.2 × 10^−20^). The top ten most significant genes are presented in Table 2, and the complete list of all genes is provided in Appendix A.

### 2.2. Clinical Sample Database Construction

Our search criteria were met by a total of 805 CRC patients’ samples from 12 datasets with available treatment information including response and raw gene expression data (Figure 1B). The detailed characteristics of all screened datasets are presented in Table 3. After normalization of the raw gene expression data and standardization between the different measurement platforms (Figure 1C), mRNA expression for a total of 19,890 genes was available. Out of 805 patients, we filtered those who had received multiple treatments: 180 patients had received bevacizumab, 221 had received irinotecan, and 438 oxaliplatin. Note that some patients had received multiple combinations of these; thus, the number of samples remaining—of patients who had received monotherapies of either 5FU or capecitabine—was reduced to 157. As our goal was to directly link the in vitro data (which was based on 5FU monotherapy) and the clinical samples, we used only these 157 samples as the validation cohort in the present study. However, the entire cohort is available for further research at the www.rocplot.com/colorectal website.

### 2.3. Validation of Significant Genes in the Clinical Sample Database

Of a total of 2484 genes previously identified in the in vitro database, 742 genes reached a statistically significant association in the clinical samples as well. The complete list of all significant genes is presented in Appendix A. The radar chart of AUC values and gene expression boxplots in resistant and sensitive samples of the strongest genes are presented in Figure 2.

### 2.4. Functional Annotation of Validated Genes

Significant genes were further examined by gene ontology biological process and molecular function overrepresentation analysis. The most significant terms were response to oxidative stress (35 genes, adjusted *p*-value: 3.90 × 10^−2^) for biological process and GTPase activity (28 genes, adjusted *p*-value: 9.79 × 10^−3^) for molecular function. The strongest significant GO categories are presented in Table 4.

### 2.5. Machine Learning Approach to Set Up an Integrated Predictive Tool

In the machine learning analysis, we used the entire combined database by employing the in vitro samples as the training set (total n = 592, nonresponder n = 294, responder n = 298), and the 5FU/capecitabine treated human samples as the test set (total n = 157, nonresponder n = 80, responder n = 77). A total of 12 genes remained in the final model and the genes that had the highest influence on the performance of the model were among the top ten genes identified in the in vitro databases, such as *SHISA4*, *SLC38A6*, *PRPF38B*, and *LAPTM4A*. Other genes with a significant effect on performance and overexpressed in the resistant phenotype include *LPP* and *FOXF2*, whereas genes overexpressed in the sensitive group were *RPL3*, *MAP2K7*, *PCF11*, *INTS7*, and *TAGAP* (Figure 3).

The accuracy of the random forest classifier in the test set was 0.745 (95% CI: 0.67–0.81), sensitivity was 0.83 (95% CI: 72–90%), and specificity was 0.66 (95% CI: 55–77%). The model correctly identified 66 out of 80 nonresponders and 56 out of 77 responders in the test set. The ROC AUC in the test was 0.74 (95% CI: 0.660–0.819, *p*-value: 2.17 × 10^−7^).

## 3. Discussion

In our study, we had two major goals. On the one hand, since the backbone of systemic chemotherapy treatment in colorectal cancer is based on fluoropyrimidines, we first attempted to identify a set of genes that have a potential role in the chemoresistance against these agents. On the other hand, we developed a machine learning-based predictor for the identification of sensitive and resistant colorectal tumors using gene expression data.

Some information is already available for the best-performing genes involved in chemoresistance identified in the in vitro database. *SHISA4* is a member of the Wnt pathway that has been previously linked to 5FU resistance in colorectal cancer [19,20]. The gene *SLC38A6* is involved in glutamine transport and the silencing of the gene resulted in the inhibition of cell cycle progression and cell viability in a hepatocellular cancer cell line [21]. The overexpression of *LAPTM4A* negatively regulates the function of human organic cation transporter 2 [22] and overexpression of the transporter is associated with improved survival in colorectal patients [23]. Remarkably, we have found that all the most important genes determined by the random forest model were also significant in our integrated database.

A strength of our study is the independence of the in vitro training set and the test set of patient samples. We have to emphasize that here we directly linked cell line data which were based on 5FU monotherapy to clinical samples with fluoropyrimidine monotherapy. To our knowledge, this is the first such study with clinically meaningful sample numbers and with a sole focus on fluoropyrimidine monotherapy.

The robustness of our results is also supported by the fact that multiple significant genes of the most important gene ontology categories also contain several markers that have been previously linked to chemoresistance. For example, silencing the *NFE2L2* gene increased 5FU sensitization in hepatocellular carcinoma cells [24], *RTN4* knockdown promoted higher cytotoxic events in paclitaxel-treated cancer cells [25], and *RND3* overexpression promoted drug resistance in gastric cell lines [26].

We have to note a limitation of our study. Although the sensitivity of the random forest model for identifying non-responder patients was 0.83, the achieved specificity was only 0.66. In other words, while the model had a high power for predicting resistance, this did not mean that the alternative estimate automatically confersi therapeutic sensitivity. A future study with an improved prediction model based on higher sample numbers could strengthen the classification sensitivity as well. Secondly, we have to note that we used tertiles of the AUDRC values for classification so that there were no cell lines with similar resistance values. In the nonresponder cell lines, the dose–response curves were markedly shifted to the right. While there could be a difference in response among these cells, the clinical utility of such high concentrations is negligible.

In summary, we identified a set of genes related to resistance against fluoropyrimidines by using gene expression profiles from cell lines and 5FU- and capecitabine-treated patients. Our results will help to rank 5FU biomarker candidates in future studies. We also constructed a machine learning model capable of linking the in vitro and in vivo models, which was able to correctly identify a high proportion of non-responder patients. By utilizing gene expression data from the primary tumor, the random forest model could be used to help therapeutic decision-making.

## 4. Materials and Methods

### 4.1. In Vitro Biomarker Discovery

To investigate the genes which may potentially influence the efficacy of 5FU therapy in vitro, we used the Genomics of Drug Sensitivity in Cancer portal (GDSC) version 1 drug screening database as a discovery dataset [27]. The preprocessing and normalization steps for generating the gene expression data tables were executed as described previously [28].

In the in vitro dataset, we used the reported area under the dose–response curve (AUDRC) values to determine the therapeutic response in 5FU-treated cell lines. We defined a cell line as a responder to the treatment if the reported AUDRC value was in the lower tertile considering all 5FU-treated cell lines. Cell lines with an AUDRC value in the upper tertile were categorized as nonresponders. Cell lines with AUDRC values in the middle tertile were excluded from the analysis (Figure 1A). The IC50 values for the cell lines are provided in Appendix A.

Gene expression across all genes among nonresponders and responders was compared using Mann–Whitney and receiver operating characteristic (ROC) tests in the R statistical environment (https://www.r-project.org/, accessed on 30 October 2022) using Bioconductor libraries (www.bioconductor.org, accessed on 30 October 2022). The significance cutoff for *p*-values was set at *p* < 0.01. The false discovery rate (FDR) was calculated using the web application www.multipletesting.com (accessed on 30 October 2022) [29], and only results with an FDR of less than 1% were accepted as significant. 

The ‘clusterProfiler version 4.0.5’ R package [30] was used to perform the functional gene annotations of the new biomarker candidates.

### 4.2. Validation in a Dataset Comprising Clinical Samples

We searched the GEO (https://www.ncbi.nlm.nih.gov/geo/, accessed on 30 October 2022) and GDC (https://portal.gdc.cancer.gov/, accessed on 30 October 2022) portals to identify datasets suitable for the analysis. During this search, the keywords “colorectal”, “cancer”, “treatment”, “response”, and “survival” were used. We considered only those publications which included raw gene expression data, information on clinical treatment, response to the treatment or survival information, and involved at least ten patients.

The integrated database comprised two Affymetrix platforms (GPL96: Affymetrix Human Genome U133A, GPL570: Affymetrix Human Genome U133 Plus 2.0), one Agilent platform (GPL6480: Whole Human Genome Microarray 4 × 44K), and the Illumina HiSeq platform.

The raw gene expression data were processed according to standard practices. For the samples measured by Affymetrix gene arrays, normalization was performed with the affy package [31], for the samples with Agilent-based chip, pre-processing was performed with the limma package [32], and samples measured by Illumina HiSeq data were processed using DESeq2 [33]. The integrated complete gene expression dataset consisting of samples from all three platforms was quantile normalized. Finally, a scaling normalization was applied to set the mean expression across all genes in each sample to 1000.

Since most samples were measured with the Affymetrix GPL570 platform, we used this platform as the basis for gene mapping between different platforms. Samples measured by other platforms were mapped to the GPL570 platform via the official gene symbols. We used JetSet [34] to select the most reliable probe set for genes measured by multiple probes. In cases where multiple probes matched a single gene, we selected the probe with the highest interquartile range [35]. 

Clinical samples were divided into responder and nonresponder groups according to the authors’ categorization using the clinical annotation of the source dataset. If the outcome of therapeutic response presented four classes, they were combined into two categories: tumors with “progression” and “stable disease” were categorized as nonresponders, whereas samples with “partial response” or “complete response” were categorized as responders. In the clinical cohort, all genes significant in the training set were analyzed for correlation with resistance.

### 4.3. Machine Learning Approach to Set Up an Integrated Predictive Tool

We aimed to combine all available genes into a single predictive algorithm. For this, we had to combine the gene expressions of cell lines and human samples into a single database. As the gene expression dynamic ranges yielded a marked difference, we performed an additional normalization using the combat command of the SVA (version: 3.40.0) R package [36]. As for the analysis of the single genes, in the model-building process, the in vitro samples were assigned to the training set and the clinical samples to the test set. For the analysis, we retained only genes upregulated in nonresponder samples if the *p*-value in the Mann–Whitney test was below 0.01 and the FDR was below 1%. Next, we applied the boruta R package as a second feature selection method to detect the most important significant genes, which were then combined into a single predictor using a random forest algorithm. The randomForest (version: 4.6.14) [37] and caret (version: 6.0.90) [38] R packages were used in the analysis. To evaluate the overall performance of the final model, we performed a receiver operating characteristic analysis and determined the area under the curve (AUC). 

## Figures and Tables

**Figure 1 ijms-24-01508-f001:**
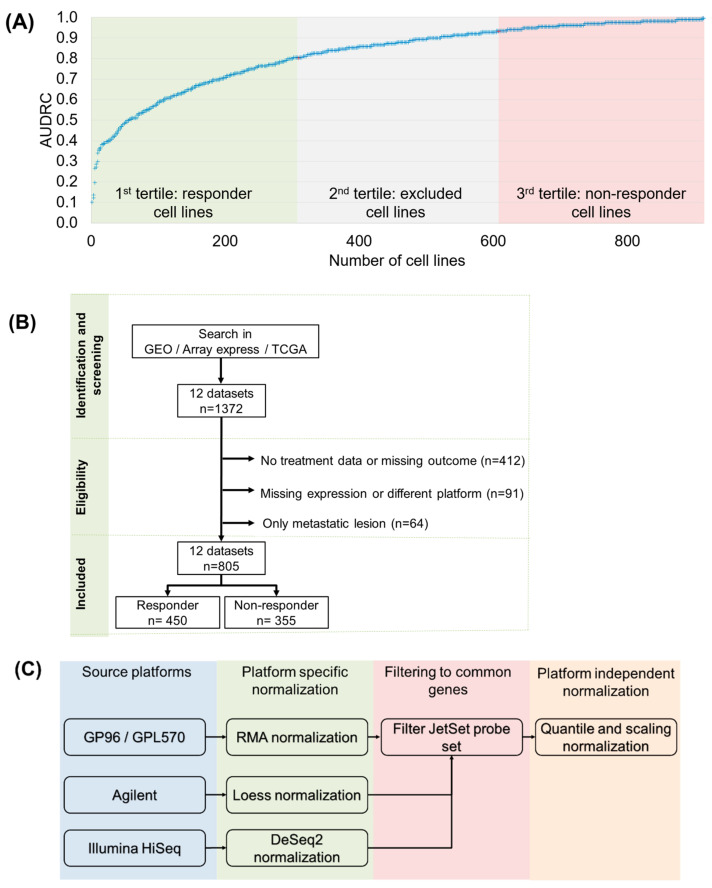
Overview of the data processing steps. Summary of response classification using the area under the dose–response curve (AUDRC) values in the 5FU-treated cell lines (**A**). Outline of the clinical database setup (**B**) and the pipeline employed for transcriptomic data pre-processing (**C**).

**Figure 2 ijms-24-01508-f002:**
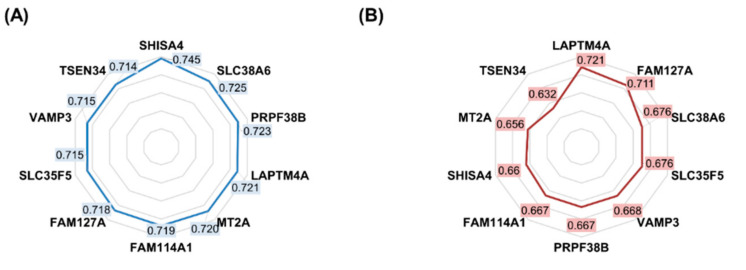
Radar chart of the most significant genes correlated with 5FU therapy response in the in vitro database (**A**) and the human samples (**B**). Boxplots of the most significant genes in both databases (**C**–**F**). The values presented in the radar chart are ROC AUC values for the specific genes.

**Figure 3 ijms-24-01508-f003:**
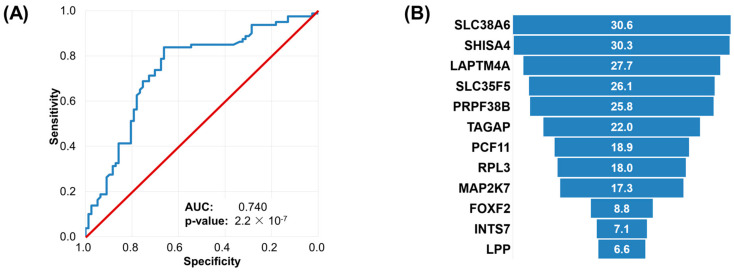
Performance of the random forest-based integrated predictor in the test set comprising of the human samples. The ROC curve of the random forest model tested in the human samples (**A**) and feature importance of the included genes determined by the random forest model (**B**). The red diagonal line depicts a random classifier model, whereas blue line displays our classifier model.

**Table 1 ijms-24-01508-t001:** The top ten 5-fluorouracil sensitive (A) and resistant (B) solid tumor cell lines in the GDSC database. A lower AUDRC (area under the dose–response curve) value indicates higher sensitivity, while a higher AUDRC value indicates greater resistance.

(A)		
Cell Line CCLE Name	Tumor Type	Normalized AUDRC
PSN1_PANCREAS	pancreas exocrine adenocarcinoma	0.352
CAL148_BREAST	breast ductal carcinoma	0.353
JHU011_UPPER_AERODIGESTIVE_TRACT	upper aerodigestive squamous carcinoma	0.39
PCI4B_UPPER_AERODIGESTIVE_TRACT	upper aerodigestive squamous carcinoma	0.423
CAL27_UPPER_AERODIGESTIVE_TRACT	upper aerodigestive squamous carcinoma	0.439
PA1_OVARY	ovary mixed germ cell	0.452
MRKNU1_BREAST	breast carcinoma	0.46
22RV1_PROSTATE	prostate adenocarcinoma	0.483
HUPT4_PANCREAS	pancreas exocrine adenocarcinoma	0.488
A2780_OVARY	ovary endometrioid adenocarcinoma	0.496
**(B)**		
**Cell Line CCLE Name**	**Tumor Type**	**Normalized AUDRC**
LN18_CENTRAL_NERVOUS_SYSTEM	glioblastoma	0.991
HEC1_ENDOMETRIUM	uterus endometrial adenocarcinoma	0.99
ASH3_THYROID	thyroid carcinoma	0.988
SCH_STOMACH	choriocarcinoma	0.988
CCFSTTG1_CENTRAL_NERVOUS_SYSTEM	astrocytoma	0.988
NCIH2444_LUNG	non-small cell lung cancer	0.988
HSC2_UPPER_AERODIGESTIVE_TRACT	upper aerodigestive squamous carcinoma	0.988
NCIH524_LUNG	small cell lung cancer	0.988
SW13_ADRENAL_CORTEX	adrenal cortex	0.987
FTC133_THYROID	thyroid carcinoma	0.987

**Table 2 ijms-24-01508-t002:** The top ten genes associated with response in the fluoropyrimidines treated samples of the in vitro database.

Gene Symbol	Approved Name	ROC AUC	Mean Expression (Nonresponder)	Mean Expression (Responder)	Mean Fold Change	Mann–Whitney U Test *p*-Value
*SHISA4*	shisa family member 4	0.745	1096.7	714.4	1.54	5.52 × 10^−25^
*SLC38A6*	solute carrier family 38 member 6	0.725	1072.0	735.3	1.46	3.07 × 10^−21^
*PRPF38B*	pre-mRNA processing factor 38B	0.723	753.0	1056.3	0.71	6.39 × 10^−21^
*LAPTM4A*	lysosomal protein transmembrane 4 alpha	0.721	1027.9	712.3	1.44	1.23 × 10^−20^
*MT2A*	metallothionein 2A	0.720	1038.1	699.6	1.48	2.34 × 10^−20^
*FAM114A1*	family with sequence similarity 114 member A1	0.719	1058.4	687.9	1.54	2.73 × 10^−20^
*FAM127A*	retrotransposon Gag-like 8C	0.718	1031.1	678.9	1.52	4.75 × 10^−20^
*SLC35F5*	solute carrier family 35 member F5	0.715	1091.4	732.7	1.49	1.17 × 10^−19^
*TSPAN4*	tetraspanin 4	0.715	1090.0	703.4	1.55	1.25 × 10^−19^
*VAMP3*	vesicle-associated membrane protein 3	0.715	1051.6	719.4	1.46	1.44 × 10^−19^

**Table 3 ijms-24-01508-t003:** Overview of datasets screened for the clinical database.

Dataset (Reference)	Platform	Sample Size	Age in Years, (Mean ± SD)	5FU/Capecitabine	Irinotecan	Bevacizumab	Oxaliplatin	Of These: 5FU/Capecitabine Monotherapy
GSE19860	Affymetrix HGU133 Plus 2.0 Array	29	-	29	-	12	29	-
GSE19862	Affymetrix HGU133 Plus 2.0 Array	14	-	-	-	14	-	-
GSE28702 [10]	Affymetrix HGU133 Plus 2.0 Array	56	-	56	-	-	56	-
GSE45404 [11]	Affymetrix HGU133 Plus 2.0 Array	42	59.9 ± 11.20	42	-	-	-	42
GSE49355 [12]	Affymetrix HGU133A Array	20	59.3 ± 7.83	20	20	-	-	-
GSE52735 [13]	Affymetrix HGU133 Plus 2.0 Array	37	-	37	-	-	37	-
GSE62080 [14]	Affymetrix HGU133 Plus 2.0 Array	21	-	21	21	-	-	-
GSE69657 [15]	Affymetrix HGU133 Plus 2.0 Array	16	53.6 ± 12.88	16	-	-	16	-
GSE72970 [16]	Affymetrix HGU133 Plus 2.0 Array	124	61.7 ± 11.39	124	88	28	40	-
GSE104645 [17]	Agilent-014850 Whole Human Genome Microarray 4x44K	167	-	140	54	83	113	-
GSE119409 [18]	Affymetrix HGU133 Plus 2.0 Array	56	56.8 ± 11.17	56	-	-	-	56
TCGA	Illumina HiSeq	223	60.7 ± 12.04	210	38	43	147	59
Total		805	60.1 ± 11.69	751	221	180	438	157

**Table 4 ijms-24-01508-t004:** Result of the functional gene annotations of the new biomarker candidates identified in in vitro and validated human samples. The table shows only the top five genes in terms. The first value is the ROC AUC in vitro, and the second value is the ROC AUC in human samples.

Class	Gene Ontology ID	Gene Ontology Term	Gene Ratio	Adjusted *p*-Value	Top Five Genes included in the Term
Biological process	GO:0006979	response to oxidative stress	35/679	3.90 × 10^−2^	*NFE2L2* (0.619; 0.714); *HNRNPD* (0.645; 0.698); *MGST1* (0.610; 0.696) *EGLN1* (0.631; 0.695); *STX4* (0.636; 0.689)
	GO:0000302	response to reactive oxygen species	22/679	4.01 × 10^−2^	*NFE2L2* (0.619; 0.714); *HNRNPD* (0.645; 0.698); *EGLN1* (0.631; 0.695); *EEF2* (0.668; 0.689); *PPP2CB* (0.659; 0.689)
	GO:1903311	regulation of mRNA metabolic process	28/679	4.01 × 10^−2^	*SF1* (0.670; 0.731); *GIGYF2* (0.624; 0.702); *HNRNPD* (0.645; 0.698); *HNRNPC* (0.621; 0.688); *THRAP3* (0.636; 0.684)
	GO:0048762	mesenchymal cell differentiation	22/679	4.01 × 10^−2^	*PDCD6* (0.624; 0.718); *TAPT1* (0.610; 0.708); *CLASP2* (0.651; 0.697); *RTN4* (0.675; 0.690); *SDCBP* (0.644; 0.680)
Molecular function	GO:0003924	GTPase activity	28/693	9.79 × 10^−3^	*RND3* (0.701; 0.726); *ARF4* (0.652; 0.703); *KRAS* (0.614; 0.694); *EEF2* (0.668; 0.689); *TUBB2A* (0.673; 0.681)
	GO:0005525	GTP binding	31/693	9.89 × 10^−3^	*RND3* (0.701; 0.726); *ARL11* (0.630; 0.703); *ARF4* (0.652; 0.703); *KRAS* (0.614; 0.694); *EEF2* (0.668; 0.689)
	GO:0032550	purine ribonucleoside binding	31/693	9.89 × 10^−3^	
	GO:0001883	purine nucleoside binding	31/693	9.89 × 10^−3^	
	GO:0032549	ribonucleoside binding	31/693	1.10 × 10^−2^	

## Data Availability

The cell line cohort and the entire human colorectal cancer cohort have been added to the ROCplot.com analysis platform.

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
