# Peer review of "New Transcriptomic Biomarkers of 5-Fluorouracil Resistance"

_ijms, 2023, doi:10.3390/ijms24021508_

Round 1

Reviewer 1 Report

An interesting study in which the authors attempted to link clinical and in vitro data and build a predictor algorithm based on deep machine learning. Based on transcription profiles, the program developed by the authors turned out to be capable of predicting the potential insensitivity of specific patients to 5-fluoropyrimidine therapy with a fairly high degree of probability. Such a program can serve as an aid in making therapeutic decisions. The expert did not see a serious flaw in the prepared manuscript, and all minor flaws can be corrected in the course of proofreading. The expert considers the study useful and worthy of consideration by the scientific community.

Author Response

We thank for the positive remarks of the reviewer.

Reviewer 2 Report

Very well written manuscript!

Line 14: “hits”, please try to use different word, you may use “signal” or any other word

Line 38: “ the adjuvant therapy for ……..CRC, please add “curatively resected” or you may write “resected”

Line 40-41: Treatment for metastatic patients is based on 5FU, oxaliplatin, and irinotecan, but may also include biological therapies targeting the VEGF or EGFR pathways.  There are more targeted therapies in addition to two you have mentioned, so please write something along the lines “may also include biological therapies such as, or write depending on genetic profiling/sidedness…. or you may change accordingly 

Line 87-88: We defined a cell line as a responder to the treatment if the reported AUDRC value was in the lower tertile considering all 5FU-treated cell lines. Is there any cut points you would like to mention for responders/nonresponders ?

Line 222-223: could you please also specify 95% CI of both sensitivity and specificity

Author Response

We thank for the positive remarks of the reviewer and have corrected and extended the manuscript at the given locations according to the suggestions. The AUDRC cutoff values are now provided in the Results section. 95% confidence intervals are displayed in page 8.

Reviewer 3 Report

The paper provided some valuable insight on the therapeutic prediction of 5FU treatment but some of the questions need to be further addressed.

1. The language of the manuscript is not well polished. Please go through the entire article and correct all the grammar errors. Professional help would be ideal in this case.

2. Please provide a representative image illustrating the dose-response curve and how AUDRC was defined in the manuscript.

3. What is the range of 5FU dose for all the referred study/dataset? The distribution of the 5FU dose in treatments could create bias on the result.

4. According to the description in Methods, cell lines with an AUDRC value in the upper tertile were categorized as nonresponders. This is questionable, as the 'nonresponder' cohort includes cell lines that could positively response to 5FU treatment. The transcriptomic profile for this group of cells is very valuable because it could explain the some side effect of 5FU treatment in the heterogenous tumor microenvironment. Please add this part of data and expand the discussion accordingly.

5. Are there any normal cells included in the model as baseline or control? Missing such a 'normal' cohort could largely affect the fidelity of the data. Please add more relevant data and expand the discussion accordingly.

Author Response

The paper provided some valuable insight on the therapeutic prediction of 5FU treatment but some of the questions need to be further addressed.

  1. The language of the manuscript is not well polished. Please go through the entire article and correct all the grammar errors. Professional help would be ideal in this case.

 The English of the manuscript has been corrected at multiple locations by a professional English corrector.

  1. Please provide a representative image illustrating the dose-response curve and how AUDRC was defined in the manuscript.

 A new figure 1A was added to clarify this issue.

  1. What is the range of 5FU dose for all the referred study/dataset? The distribution of the 5FU dose in treatments could create bias on the result.

 The results section now includes additional details about the 5FU concentrations.

  1. According to the description in Methods, cell lines with an AUDRC value in the upper tertile were categorized as nonresponders. This is questionable, as the 'nonresponder' cohort includes cell lines that could positively response to 5FU treatment. The transcriptomic profile for this group of cells is very valuable because it could explain the some side effect of 5FU treatment in the heterogenous tumor microenvironment. Please add this part of data and expand the discussion accordingly.

We have especially used tertiles of the AUDRC values so that there were no cell lines with similar resistance values. In the nonresponder cell lines the dose-response curves are markedly shifted to the right - while there are also differences in response among these cells, the clinical utility of such high concentrations is negligible. We have extended the limitations part in the Discussion section to discuss this issue.

  1. Are there any normal cells included in the model as baseline or control? Missing such a 'normal' cohort could largely affect the fidelity of the data. Please add more relevant data and expand the discussion accordingly.

We investigated cancer cell lines treated with 5FU. We have checked the GEO database and we could not find datasets for normal cell lines with available transcriptomic data where 5FU treatment was administered.

Round 2

Reviewer 3 Report

1. Please check the label in Figure 1A and label the non-responders.

2. A figure showing the 5FU concentration distribution for all the dataset in the study should be included.

Author Response

  • We improved Figure 1A increase clarity.
  • We added a new Supplemental Figure 1 with the ranked IC50 values for all cell lines (detailed dosage information was not available for the human samples)
  • A professional English corrector re-examined the entire manuscript. Please see the changed tracked by yellow background.